# External Validation of Risk Scores for Predicting Venous Thromboembolism in Ambulatory Patients with Lung Cancer

**DOI:** 10.3390/cancers16183165

**Published:** 2024-09-15

**Authors:** Ann-Rong Yan, Desmond Yip, Gregory M. Peterson, Indira Samarawickrema, Mark Naunton, Phillip Newman, Reza Mortazavi

**Affiliations:** 1School of Health Sciences, Faculty of Health, University of Canberra, Bruce, ACT 2617, Australia; ann-rong.yan@canberra.edu.au (A.-R.Y.); mark.naunton@canberra.edu.au (M.N.); 2Department of Medical Oncology, The Canberra Hospital, Garran, ACT 2605, Australia; desmond.yip@anu.edu.au; 3ANU School of Medicine and Psychology, Australian National University, Canberra, ACT 2601, Australia; 4College of Health and Medicine, University of Tasmania, Hobart, TAS 7005, Australia; g.peterson@utas.edu.au; 5School of Nursing, Midwifery and Public Health, Faculty of Health, University of Canberra, Bruce, ACT 2617, Australia; infoindira9@gmail.com; 6Strategy Coaching and Research Consulting Pty Ltd., Canberra, ACT 2606, Australia; 7Research Institute for Sport and Exercise, Faculty of Health, University of Canberra, Bruce, ACT 2617, Australia; phillip.newman@canberra.edu.au

**Keywords:** lung cancer, venous thromboembolism, risk scores, risk factors, discriminatory capability, validation, thromboprophylaxis

## Abstract

**Simple Summary:**

People living with cancer are at a higher risk of developing a venous thromboembolism (VTE). A VTE may interrupt anticancer therapy, which increases the risk of mortality. Cancer treatment guidelines recommend the use of preventive anticoagulants only in those at high risk of a VTE, identified by a valid risk assessment tool, such as the Khorana score. However, the Khorana score at its original cut-off value of 3 points has a low sensitivity and discriminatory capability in lung cancer patients, and the updated Khorana score with a 2-point cut-off value lacks validation in this population. Other risk scores, such as the PROTECHT, CONKO, and COMPASS-CAT scores, have not been validated in large lung cancer cohorts. The aim of this study was to evaluate these four existing risk scores in this patient population. We validated the Khorana score with a cut-off value of 2 points and the CONKO score with a cut-off value of 2 points.

**Abstract:**

Background: The purpose of this study was to evaluate the discriminatory capability of the Khorana, PROTECHT, CONKO, and COMPASS-CAT scores in ambulatory patients with lung cancer. Methods: This retrospective cohort study included 591 patients with newly diagnosed lung cancer. A symptomatic or incidental VTE occurred in 108 patients. Results: The Khorana score at a 2-point threshold had a discriminatory capability with an odds ratio (OR) of 1.80 and an AUC of 0.57 for 6 months, and an OR of 1.51 and an AUC of 0.55 for 12 months. The CONKO score at a 2-point threshold had a stronger discriminatory capability for both 6 months and 12 months with ORs of 3.00 and 2.13, and AUCs of 0.63 and 0.59, respectively. Additionally, higher white blood cell counts, higher neutrophil counts, hypoalbuminaemia, and not undergoing lung surgery were related to VTE occurrence (*p* < 0.05). Conclusions: The Khorana score with the 2-point threshold was validated in ambulatory patients with lung cancer, with the results indicating a decline in its discriminatory capability over time (at 12 months vs. 6 months from diagnosis). The CONKO score at the original 2-point threshold showed a stronger discriminatory capability but further validation with a larger sample size is recommended. The identified predictors should be further investigated in future research.

## 1. Introduction

Patients living with cancer, compared with the general population, are at higher risk of developing a venous thromboembolism (VTE) [1]. The annual incidence of VTE in patients diagnosed with lung cancer is around 11% (95% CI 8%–14%) [2]. This is much higher than the annual incidence of VTE among the general population, which is 0.1% [1]. Developing a VTE tripled the risk of death during a median follow-up of 23.1 months in this group of patients (HR 3.09, 95% CI 1.07, 4.60) [3]. Specifically, having a VTE predicted death in patients with lung cancer for both non-small-cell lung cancer (NSCLC) and small-cell lung cancer (SCLC), with a hazard ratio (HR) of 2.3 (95% CI 2.2, 2.4) and 1.5 (95% CI 1.3, 1.7), respectively [4]. Apart from an increased risk of mortality, having a VTE may require clinicians to temporarily stop the anticancer therapy, which can lead to the progression of the cancer and an increased risk of mortality [3].

Several randomised controlled trials have demonstrated that pharmacological thromboprophylaxis is effective in reducing the risk of VTE in patients with cancer [5,6,7]. In the TARGET-TP trial, enoxaparin decreased the risk of VTE by 69% (95% CI 30%, 85%), from 23% to 8%, in patients with lung or gastrointestinal cancer who were at high risk at 6 months [6]. In the AVERT trial, apixaban, a direct-acting oral anticoagulant (DOAC), significantly lowered the VTE rate by 59% (95% CI 35%, 74%) from 10.2% to 4.2% at 6 months in cancer patients at intermediate-to-high VTE risk [5], while in the CASSINI trial, rivaroxaban reduced the VTE rate by 60% (95% CI 20%, 80%), from 6.4% to 2.6%, during the cancer intervention period [7].

Despite the effectiveness of anticoagulants in preventing a VTE, the associated risk of bleeding cannot be ignored [8]. Accordingly, the American Society of Clinical Oncology (ASCO) guidelines [9] recommend the use of primary thromboprophylaxis only for cancer patients whose VTE risk is deemed high using the Khorana score [10]. However, the Khorana score at the original cut-off value of 3 points (to indicate a high VTE risk) was reported to have a low sensitivity [11] and discriminatory capability in patients with lung cancer [12,13]. Consequently, in the updated ASCO Guideline, a cut-off value of 2 points for the Khorana score was adopted for indicating a high VTE risk [9].

Our previous systematic review showed that the Khorana score with the cut-off value of 2 or 3 points, the PROTECHT score with a cut-off of 3 points [14], and the CONKO score with a cut-off value of 3 points [15] were not capable of stratifying lung cancer patients at a high or low VTE risk, whereas the COMPASS-CAT score [16] with a cut-off value of 7 points was useful for this purpose [2]. However, 2 of the 3 studies that were included for the meta-analysis of the performance of the COMPASS-CAT score had small sample sizes of 118 [17] and 150 [18]. As a result, it was concluded that a validation of this score with a larger sample size was needed, and that the observed optimal cut-off value of 11 points was also in need of external validation. In addition, the updated Khorana, PROTECHT, and CONKO scores needed to be externally validated.

The aim of this cohort study was to externally validate the Khorana, PROTECHT, CONKO, and COMPASS-CAT scores for identifying high VTE risk in patients with lung cancer. We also aimed to identify risk factors which had not been included in the existing risk scores.

## 2. Materials and Methods

### 2.1. Study Design

A single-centre retrospective cohort study was conducted in Canberra, Australia on outpatients with newly diagnosed primary lung cancer, registered in the database of Canberra Region Cancer Centre (CRCC) named Charm HealthTM (Charm Health Pty Ltd., Brisbane, Australia), from 1 January 2012 to 11 November 2022. Consecutive adult (≥18 years of age) ambulatory outpatients who had a histological diagnosis of primary lung cancer during the study period were screened for inclusion in this study. Patients were excluded if they had no records of the date of diagnosis of lung cancer or were diagnosed with lung cancer before 1 January 2012 or after 11 November 2022, if they had a VTE at the time of diagnosis of lung cancer or within 3 months prior to it, or if they were hospitalised during the follow-up period.

The primary outcome was a symptomatic or incidental VTE diagnosed within 6 months or 12 months from the diagnosis of primary lung cancer. VTE diagnoses were confirmed through medical imaging reports of Doppler ultrasound, CT pulmonary angiograms, or CT thorax conducted for staging or restaging purposes. The included patients were followed up from the diagnosis of primary lung cancer until reaching one of the following endpoints: (1) first VTE occurrence, (2) the end of the 12-month follow-up, (3) loss to follow-up, or (4) death.

### 2.2. Study Variables

The medical records of the participants were reviewed. The collected data included date of birth, sex, height, weight, smoking status, cancer type, date of diagnosis, stage and grade of cancer, metastatic sites, radiotherapy, prescriptions for chemotherapy, immunotherapy, anticoagulants, antiplatelets, oestrogen, raloxifene, corticosteroids, antipsychotics, erythropoietin, fenofibrate and statins, the use of a central venous catheter, resection surgery, history of VTE, peripheral artery disease, ischaemic stroke, coronary artery disease, hypertension, hyperlipidaemia, diabetes, obesity, and baseline biomarkers, such as full blood count, coagulation profile, and routine biochemistry. The times from diagnosis of lung cancer to the diagnosis of the first VTE (if any) and/or to death were recorded. Individual patients’ scores for the Khorana, PROTECHT, CONKO, and COMPASS-CAT scores were calculated where full data were available (the elements of each risk score are shown in Appendix A).

### 2.3. Statistical Analysis 

Mean ± SD were used to describe normally distributed continuous variables, with median and interquartile range for non-normally distributed continuous data. Chi-square tests, two-sample *t*-tests, and Mann–Whitney U tests were used, as appropriate, to compare groups. No statistical test was conducted for variables with a prevalence less than 1%. The Hosmer–Lemeshow test was used to evaluate the goodness-of-fit. Sensitivity (the probability of high risk in those patients having a VTE), specificity (the probability of low risk in those without a VTE), positive predictive value (PPV; the probability of high risk in those patients identified to be at high risk), and negative predictive value (NPV; the probability of no VTE in those patients identified to be at low risk) for VTE development were estimated. The cumulative area under receiver operating characteristic curve (AUC) was estimated to evaluate the discriminatory capability of the risk scores. Kaplan–Meier curves were constructed, and the log-rank test was conducted to demonstrate the difference in VTE probability between risk groups stratified by the risk scores. The data were analysed with Stata software version SE 18 (StataCorp, College Station, TX, USA). For hypothesis testing, a *p* value < 0.05 was considered statistically significant.

## 3. Results

### 3.1. Patient Characteristics 

A total of 847 patients were screened, of whom 681 patients met the inclusion criteria and 166 were excluded for the following reasons: diagnosed with secondary or unknown primary lung cancer (*n*= 26); diagnosed with lung cancer before 2012 (*n* = 105); had a VTE at lung cancer diagnosis or within 3 months prior to it (n = 30); or were hospitalised during the follow-up period (*n* = 5) (Figure 1). As 90 patients were lost to follow-up, 591 patients were included for the analysis. Of the included patients, 86 had the first VTE within 6 months after the diagnosis of lung cancer, and a further 22 developed a VTE between 6 and 12 months following lung cancer diagnosis. Of the total of 108 (18.3% of 591) VTE-diagnosed patients, there were 24 deep vein thrombosis (DVT) patients and 70 pulmonary embolism (PE) patients. Nine patients were diagnosed with both DVT and PE, and five patients had other types of VTE, including jugular vein thrombus, superior mesenteric vein thrombus, or superior vena cava obstruction. The characteristics of patients who developed or did not develop a VTE within 12 months of lung cancer diagnosis are summarised in Table 1. Additionally, there were four cases diagnosed with peripheral artery disease (PAD), nine with coronary artery disease (CAD), eight with ischaemic stroke, and three with both CAD and ischaemic stroke during the follow-up period. In total, 236 (40%) people died by the end of the 12-month follow-up. The mortality rate among patients who developed a VTE was significantly higher than that among those who did not develop a VTE (49% vs. 38%, respectively; *p* = 0.05) (Table 2).

### 3.2. Performance of Risk Scores

The Khorana score at a 2-point threshold for high VTE risk had a low discriminatory capability, with an OR of 1.80 (95% CI 1.11, 2.88) and an AUC of 0.57 (95% CI 0.51, 0.63) for 6 months, and an OR of 1.51 (85% CI 0.98, 2.32) and an AUC of 0.55 (95% CI 0.50, 0.60) for 12 months. By contrast, the CONKO score at a 2-point threshold had better discriminatory capability for both 6 months and 12 months, with odds ratios of 3.00 (95% CI 1.52, 5.91) and 2.13 (95% CI 1.19, 3.81), and AUCs of 0.63 (95% CI 0.56, 0.69) and 0.59 (95% CI 0.52, 0.66), respectively. The Khorana score at a 3-point threshold and the CONKO score at a 3-point threshold did not have a satisfactory discriminatory capability (Table 3). Neither the PROTECHT score nor the COMPASS-CAT score were able to discriminate between high- and low-risk patients (Table 3). According to the Hosmer–Lemeshow tests, the *p* values were greater than 0.05, indicating that each of the risk scores was well calibrated (Table 3).

The Kaplan–Meier curves showed that the high-risk groups stratified by the Khorana score at a 2-point threshold and the CONKO score at a 2-point threshold exhibited significantly higher probability of a VTE (*p* values of 0.02 and 0.003, respectively) (Figure 2).

In the sensitivity analysis of the CONKO score at a 2-point cut-off, 105 patients who had missing data in ECOG PS but already scored 2 points for the rest of the elements in the CONKO score were included. The results were consistent with a significant association and showed a similar trend over time. The CONKO score at a 2-point threshold for high VTE risk had an intermediate discriminatory capability with an OR of 2.92 (95% CI 1.53, 5.57) and an AUC of 0.61 (95% CI 0.55, 0.66) for 6 months, and an OR of 2.18 (85% CI 1.26, 3.76) and an AUC of 0.58 (95% CI 0.53, 0.63) for 12 months.

### 3.3. Risk Factors

Compared to patients without a VTE, patients with a VTE more commonly had higher median WBC counts (9.9 × 10^9^/L vs. 8.8 × 10^9^/L, *p* = 0.009), leucocytosis (37% vs. 26%, *p* = 0.02), higher median neutrophil count (6.70 × 10^9^/L vs. 6.01 × 10^9^/L, *p* = 0.01), and rate of hypoalbuminaemia (16% vs. 9%, *p* = 0.02), whilst a smaller proportion of patients in the VTE group underwent surgery as a treatment (6% vs. 18%, *p* = 0.002) (Table 1).

## 4. Discussion

This retrospective study validated the Khorana score at the 2-point cut-off value for predicting VTE in ambulatory patients with lung cancer and provided evidence to support the ASCO guidelines’ recommendation of this cut-off value over the original 3-point cut-off value for the score [9,10]. This cohort study also demonstrated the discriminatory capacity of the CONKO score for VTE prediction within 12 months after the diagnosis of lung cancer. This finding could be of clinical usefulness in patients with lung cancer. When full blood counts are not tested and the Khorana score cannot be calculated, the CONKO score may be used as an alternative assessment to identify high VTE risk if ECOG PS is 2 points or higher.

The CONKO score is a modification of the Khorana score with the replacement of BMI ≥ 35 kg/m^2^ by ECOG PS ≥ 2 points [15], and the lower the ECOG PS is, the lower the CONKO score for individual patients becomes. In this study, no association was found between ECOG PS and VTE, which is consistent with the literature [3]; however, patients eligible for chemotherapy or lung surgery usually have a lower ECOG PS [17]. In our study, a CONKO score of less than 2 points and undergoing lung surgery were both consistently associated with a decreased VTE risk. The latter finding may seem unexpected and contradictory given that surgical operation is a risk factors for VTE in general; however, this may be justified by the fact that the main cause of a cancer-associated VTE is the cancer itself, therefore the removal of tumours in this cohort ameliorated the patients’ hypercoagulable conditions. This view is consistent with the report by Falanga et al. [19]. In other words, changes in tumour burden over time (due to remission, recurrence or progression) could change the individuals’ VTE risk. This further supports the importance of reassessing VTE risk at different stages of the disease course.

Notably, the discriminatory capabilities of both the Khorana score and CONKO score were less at 12 months compared with 6 months, which may be an indication of the unsuitability of using baseline patient data to assess the risk of a VTE at 12 months following the diagnosis of lung cancer. A similar trend of decrease in score accuracy after 6 months was also reported in a study of heterogenous groups of cancer patients [20], suggesting the necessity of reassessing VTE risk every few months. Of the five elements in the Khorana score, three blood markers, i.e., haemoglobin, white cell count, and platelet count, are more likely to change over time than the other two elements (cancer type and obesity). This may indicate the need for future research on the possible association between the longitudinal changes in biomarkers and VTE risk [21].

This study found that the COMPASS-CAT score lacked the discriminatory capability for predicting a VTE, which is contradictory to the results of our systematic review, where the meta-analysis showed a significant association between the predicted high risk stratified with a COMPASS-CAT score ≥7 points and VTE occurrence (RR 4.68, 95% CI 1.05–20.80) [2]. Furthermore, the optimal positivity threshold of 11 points demonstrated in the previous study [17] did not improve the performance of the COMPASS-CAT score in this study. The possible explanation for our finding might be the narrower distributions in some predictors of COMPASS-CAT score. First, anthracyclines were not used in this cohort; secondly, the use of anti-hormone therapy was rare, with the prevalence being less than 2% (nine patients); thirdly, all the included participants had newly diagnosed lung cancer and the study followed up from the time of diagnosis. As a result, these predictors scored almost the same for every single individual in this study cohort, and using data with a smaller case mix for model validation tends to show lower discriminative ability than those using data sources with more broadly distributed predictors [22].

Although the study showed that the Khorana score and the CONKO score were associated with VTE occurrence, the AUCs were low to moderate. Thus, the models’ accuracy need to be improvement. On the other hand, all the risk scores had high negative predictive values, which suggests that patients with low-risk scores could be spared from pharmacological thromboprophylaxis unless there are other risk factors for VTE which are not included in the scores mentioned [23].

Apart from the cancer-related risk factors, some biomarkers showed an association with VTE risk, e.g., a white cell count greater than 11 × 10^9^/L, which is one of the predictors included in the Khorana score. Having a low albumin level was strongly associated with an increased VTE risk. The latter finding was also observed in the Vienna Cancer and Thrombosis Study (CATS) of a heterogenous group of cancer patients [24]. Therefore, a low serum albumin level should be further investigated as a potential candidate predictor for new VTE risk assessment models or for updating the current models.

Our study has some limitations. The main limitation is its retrospective design. Some variables that were not routinely collected or recorded had a significant number of missing values. For example, 218 out of 591 (37%) patients did not have a record of ECOG PS score. As a result, the validation of the CONKO score had a much smaller sample size (n = 370) with only 60 VTE events. Although the sensitivity analysis of 475 patients with 83 VTE events showed the same degree of significance, a low number of events of interest (i.e., less than 100 for external validation) is considered as a bias according to the Prediction model Risk Of Bias ASessment Tool (PROBAST - Version of 15/05/2019) [22]. Therefore, the discriminatory capability of the CONKO score needs to be validated in a larger cohort.

## 5. Conclusions

Our study validates the Khorana score with the updated 2-point threshold in patients with lung cancer, but it also suggests a need for frequent reassessment of patients’ VTE risk over time because of the observed decline in the discriminatory capability of the risk score at 12 months post-cancer diagnosis compared with 6 months post-cancer diagnosis. Our study also showed the CONKO score at the 2-point threshold has a strong discriminatory capability, but further validation with prospective studies and larger samples is recommended. In addition, our study identified additional risk factors which might be useful for developing new risk scores or updating current risk scores in future research.

## Figures and Tables

**Figure 1 cancers-16-03165-f001:**
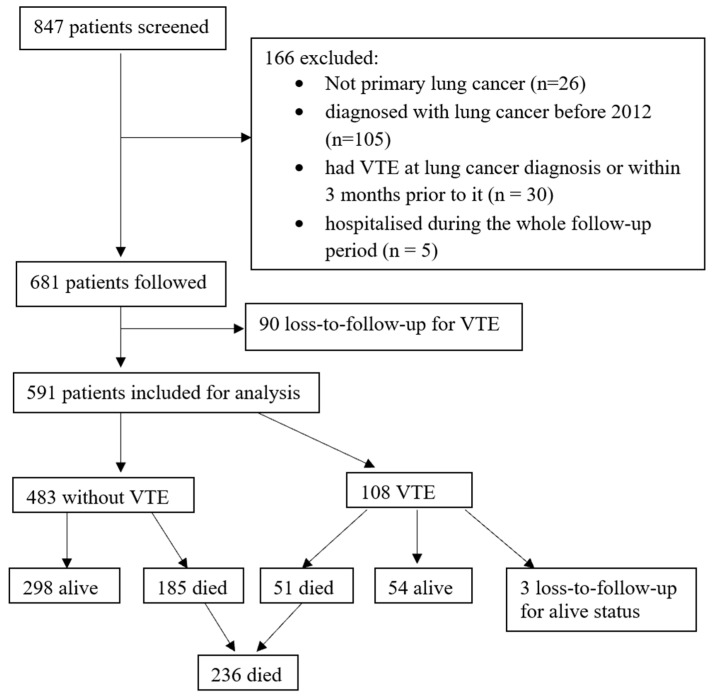
Participant selection and outcomes at 12 months after the diagnosis of lung cancer.

**Figure 2 cancers-16-03165-f002:**
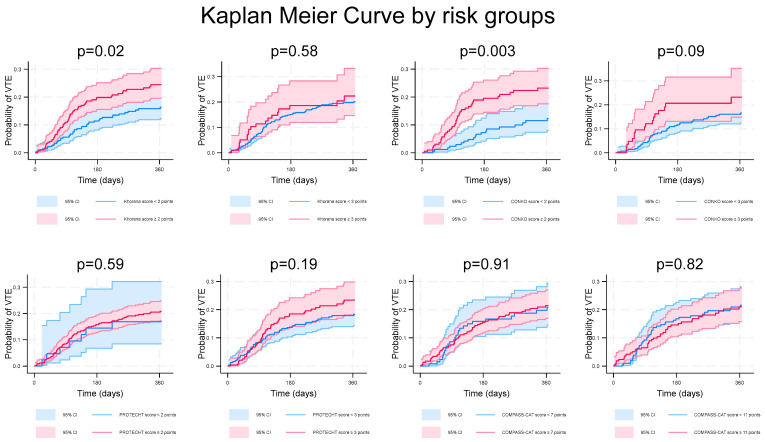
Kaplan–Meier curve of VTE probability in patients at high risk or low risk stratified by the risk scores.

**Table 1 cancers-16-03165-t001:** Characteristics of patients who developed and did not develop a VTE within 12 months of lung cancer diagnosis.

Variables	All (n = 591)	VTE (n = 108)	No VTE (n = 483)	*p* Value
Patient-related risk factors				
Age at diagnosis in years, median (p25–p75)	67 (58–74)	68 (61–74)	67 (58–74)	0.22
Male, n (%)	316 (53)	58 (54)	258 (53)	0.96
Indigenous, n (%)	20 (3)	6 (6)	14 (3)	0.17
Ever smoker, n (%)	471 (80)	88 (83)	383 (80)	0.45
Body Mass Index in kg/m^2^, median (p25–p75)	25.5 (21.7–29.9)	25.7 (21.5–29.5)	25.5 (21.9–29.9)	0.97
ECOG PS, median (p25–p75)	1 (0–1)	1 (0–1)	1 (0–1)	0.30
Obesity, n (%)	140 (24)	25 (24)	115 (24)	0.88
Hypertension, n (%)	271 (46)	48 (44)	223 (46)	0.75
Hypercholesterolaemia/hyperlipidaemia/dyslipidaemia, n (%)	196 (33)	36 (33)	160 (33)	0.97
Diabetes, n (%)	95 (16)	17 (16)	78 (16)	0.92
Peripheral artery disease, n (%)	23 (4)	2 (2)	21 (4)	0.23
Coronary artery disease, n (%)				0.44
No myocardial infarction	38 (6)	5 (5)	33 (7)	
Had myocardial infarction ^#^	45 (8)	6 (6)	39 (8)	
Ischaemic stroke, n (%)	34 (6)	5 (5)	29 (6)	0.58
Haemorrhagic stroke, n (%)	1 (<1)	0 (0)	1 (<1)	--
Heart failure, n (%)	16 (3)	2 (2)	14 (3)	0.55
Atrial fibrillation, n (%)	73 (12)	11 (10)	62 (13)	0.45
Chronic liver disease, n (%)	8 (1)	0 (0)	8 (2)	0.18
Chronic kidney disease, n (%)	28 (5)	4 (4)	24 (5)	0.58
Chronic obstruction pulmonary disease, n (%)	189 (32)	31 (29)	158 (33)	0.42
History of VTE, n (%)	26 (4)	7 (6)	19 (4)	0.24
Number of comorbidities *, median (p25–p75)	1 (0–2)	1 (0–2)	1 (0–2)	0.65
Medications during follow-up				
Anticoagulants, n (%)	75 (13)	9 (8)	66 (14)	0.13
Antiplatelets, n (%)	125 (21)	19 (18)	106 (22)	0.32
Selective estrogen receptor modulators, n (%)	6 (1)	1 (<1)	5 (1)	--
Hormone replacement therapy, n (%)	9 (2)	4 (4)	5 (1)	0.04
Corticosteroids, n (%)	262 (44)	50 (46)	212 (44)	0.65
Antipsychotics, n (%)	172 (29)	26 (24)	146 (30)	0.20
Erythropoietin, n (%)	43 (7)	4 (4)	39 (8)	0.11
Statins, n (%)	187 (32)	33 (31)	154 (32)	0.79
Proton-pump inhibitors, n (%)	226 (38)	48 (44)	178 (37)	0.14
Cancer-related risk factors				
Non-small-cell lung cancer (NSCLC), n (%)	477 (81)	91 (84)	386 (80)	0.32
Poorly/undifferentiated, n (%)	204 (67)	34 (74)	170 (66)	0.27
Advanced stage, n (%)	389 (86)	78 (92)	311 (85)	0.11
Metastasis, n (%)	439 (87)	86 (91)	353 (86)	0.25
Chemotherapy, n (%)	486 (82)	82 (76)	404 (84)	0.06
Etoposide, n (%)	115 (19)	18 (17)	97 (20)	0.42
Platinum, n (%)	462 (78)	78 (72)	384 (80)	0.10
Carboplatin, n (%)	356 (60)	63 (58)	293 (61)	0.66
Cisplatin, n (%)	117 (20)	16 (15)	101 (21)	0.15
Gemcitabine, n (%)	76 (13)	16 (15)	60 (12)	0.50
Vinorelbine, n (%)	136 (23)	21 (19)	115 (24)	0.33
Docetaxel, n (%)	13(2)	5 (5)	8 (2)	0.06
Paclitaxel, n (%)	60 (10)	8 (7)	52 (11)	0.30
Pemetrexed, n (%)	98 (17)	16 (15)	82 (17)	0.59
Radiotherapy, n (%)	287 (49)	50 (46)	237 (49)	0.58
Target therapy, n (%)	24 (4)	4 (4)	20 (4)	0.84
Immune checkpoint inhibitors, n (%)	214 (36)	32 (30)	182 (38)	0.13
Hospitalisation within 3 months prior to the diagnosis of lung cancer, n (%)	168 (28)	27 (25)	141 (29)	0.38
Central venous catheter, n (%)	39 (7)	10 (10)	29 (6)	0.24
Resection (lung surgery as cancer treatment), n (%)	96 (16)	7 (6)	89 (18)	0.002
Biomarkers				
Haemoglobin in g/L, median (p25–p75)	134 (121–144)	135 (119–144)	134 (123–145)	0.54
Haemoglobin < 100 g/L, n (%)	26 (4)	8 (7)	18 (4)	0.09
White cell count in 10^9^/L, median (p25–p75)	9.0 (7.3–11.3)	9.9 (7.9–12.6)	8.8 (7.3–11.2)	0.009
White cell count ≥ 11 × 10^9^/L, n (%)	166 (28)	40 (37)	126 (26)	0.02
Platelet count in 10^9^/L, median (p25–p75)	287 (226–358)	296 (246–365)	286 (224–354)	0.17
Platelet count ≥ 350 × 10^9^/L, n (%)	157 (27)	31 (29)	126 (26)	0.57
Red cell distribution width in %, median (p25–p75)	14.0 (13.3–14.9)	14.2 (13.3–14.9)	13.9 (13.3–14.9)	0.47
Neutrophil count in 10^9^/L, median (p25–p75)	6.11 (4.67–8.42)	6.70 (5.10–9.94)	6.01 (4.60–8.23)	0.01
Lymphocyte count in 10^9^/L, median (p25–p75)	1.58 (1.17–2.08)	1.63 (1.22–2.15)	1.57 (1.17–2.06)	0.53
Monocyte count in 10^9^/L, median (p25–p75)	0.73 (0.57–0.95)	0.73 (0.60–1.00)	0.73 (0.55–0.93)	0.11
Platelet to lymphocyte ratio (PLR), median (p25–p75)	182 (126–264)	182 (127–269)	182 (126–262)	0.92
Neutrophil to lymphocyte ratio (NLR), median (p25–p75)	3.77 (2.48–6.31)	4.28 (2.82–6.25)	3.69 (2.43–6.32)	0.20
Abnormal PT, n (%)	25 (6)	5 (6)	20 (6)	0.74
Abnormal INR, n (%)	14 (3)	0 (0)	14 (4)	0.08
Abnormal aPTT, n (%)	26 (6)	5 (6)	21 (6)	0.82
Low eGFR (<60 mL/min/1.73m^2^), n (%)	65 (11)	13 (12)	52 (11)	0.67
Total bilirubin above normal range, n (%)	24 (4)	3 (3)	21 (4)	0.46
Alanine aminotransferase above normal range, n (%)	87 (15)	16 (15)	71 (15)	0.95
Alkaline phosphatase above normal range, n (%)	160 (27)	32 (30)	128 (27)	0.48
Gamma-glutamyl transferase above normal range, n (%)	195 (33)	38 (36)	157 (33)	0.55
Albumin below normal range, n (%)	57 (10)	17 (16)	40 (8)	0.02
Corrected total calcium above normal range, n (%)	69 (12)	13 (13)	56 (12)	0.84
C-reactive protein in mg/L, median (p25–p75)	28 (8–69)	24 (10–52)	32 (7–70)	0.90

p25: 25th percentile; p75: 75th percentile; ECOG PS: Eastern Cooperative Oncology Group performance status; PT: prothrombin time; INR: international normalised ratio; aPTT: activated partial thromboplastin time; # including ST-elevation myocardial infarction (MI) and non-ST-elevation MI. * Comorbidities include peripheral artery disease, ischaemic stroke, coronary artery disease, hypercholesterolaemia/hyperlipidaemia/dyslipidaemia, hypertension, diabetes, obesity.

**Table 2 cancers-16-03165-t002:** Clinical outcomes and VTE risk scores of patients who developed and did not develop a VTE within 12 months of lung cancer diagnosis.

	All (n = 591)	VTE (n = 108)	No VTE (n = 483)	*p* Value
Clinical outcomes				
Ischaemic stroke, n (%)	11 (2)	4 (4)	7 (1)	0.12
Peripheral artery disease, n (%)	4 (<1)	1 (<1)	3 (<1)	--
Coronary artery disease, n (%)	12 (2)	4 (4)	8 (2)	0.16
Death within one year since cancer diagnosis, n (%)	236 (40)	51 (49)	185 (38)	0.05
Risk scores				
Khorana score (n = 570), median (p25–p75)	2 (1–2)	2 (1–2)	2 (1–2)	0.13
Khorana score ≥ 2 points, n (%)	298 (52)	63 (61)	235 (50)	0.06
Khorana score ≥ 3 points, n (%)	101 (18)	19 (18)	82 (18)	0.87
PROTECHT score (n = 570), median (p25–p75)	2 (2–3)	2 (2–3)	2 (2–3)	0.49
PROTECHT score ≥ 2 points, n (%)	526 (92)	97 (93)	429 (92)	0.68
PROTECHT score ≥ 3 points, n (%)	248 (44)	50 (48)	198 (42)	0.30
CONKO score (n = 370), median (p25–p75)	2 (1–2)	2 (1–3)	2 (1–2)	0.02
CONKO score ≥ 2 points, n (%)	197 (53)	41 (68)	156 (50)	0.01
CONKO score ≥ 3 points, n (%)	85 (23)	17 (28)	68 (22)	0.28
COMPASS-CAT score (n = 429), median (p25–p75)	11 (6–12)	9 (6–11)	11 (6–13)	0.84
COMPASS-CAT score ≥ 7, n (%)	298 (69)	57 (70)	241 (69)	0.99
COMPASS-CAT score ≥ 11, n (%)	219 (51)	40 (49)	179 (52)	0.65

p25: 25th percentile; p75: 75th percentile.

**Table 3 cancers-16-03165-t003:** Performance of VTE risk scores at 6 months and 12 months, respectively, since lung cancer diagnosis.

		**VTE Prevalence %** **High Risk vs. Low Risk,** ***p* Value**	**Sensitivity** **(%, 95% CI)**	**Specificity** **(%, 95% CI)**	**Positive Predictive Value (%, 95% CI)**	**Negative Predictive Value (%, 95% CI)**	**AUC** **(95% CI)**	**Odds Ratio** **(95% CI)**	**Hosmer-Lemeshow Test,** ***p* value**
6 months	Khorana score Threshold of 2 points Threshold of 3 points	18.1% vs. 11.0%, *p* = 0.02 16.8% vs. 14.3%, *p* = 0.51	64.3 (53.1, 74.4) 20.2 (12.3, 30.4)	49.8 (45.3, 54.3) 82.7 (79.1, 86.0)	17.4 (13.6, 21.8) 16.8 (10.1, 25.6)	89.5 (84.7, 93.3) 85.7 (82.2, 88.8)	0.57 (0.51, 0.63) 0.52 (0.47, 0.56)	1.80 (1.11, 2.88) 1.21 (0.68, 2.16)	0.09
	PROTECHET score Threshold of 2 points Threshold of 3 points	14.8% vs. 13.6%, *p* = 0.83 16.9% vs. 13.0%, *p* = 0.19	92.9 (85.1, 97.3) 50.0 (38.9, 61.1)	7.8 (5.6, 10.6) 57.6 (53.1, 62.1)	14.7 (11.8, 18.0) 16.9 (12.5, 22.2)	84.8 (68.1, 94.9) 87.0 (82.8, 90.4)	0.50 (0.47, 0.53) 0.54 (0.48, 0.60)	1.10 (0.46, 2.63) 1.36 (0.86, 2.16)	0.12
	CONKO score Threshold of 2 points Threshold of 3 points	18.3% vs. 6.9%, *p* = 0.001 18.8% vs. 11.2%, *p* = 0.07	75.0 (60.4, 86.4) 33.3 (20.4, 48.4)	50.0 (44.4, 55.6) 78.6 (73.7, 82.9)	18.3 (13.1, 24.4) 18.8 (11.2, 28.8)	93.1 (88.2, 96.4) 88.8 (84.5, 92.2)	0.63 (0.56, 0.69) 0.56 (0.49, 0.63)	3.00 (1.52, 5.91) 1.83 (0.96, 3.51)	0.12
	COMPASS-CAT score Threshold of 7 points Threshold of 11 points	14.8% vs. 15.3%, *p* = 0.89 13.7% vs. 16.2%, *p* = 0.47	68.8 (55.9, 79.8) 46.9 (34.3, 59.8)	30.4 (25.7, 35.4) 48.2 (43.0, 53.5)	14.8 (10.9, 19.3) 13.7 (9.4, 19.0)	84.7 (77.4, 90.4) 83.8 (78.1, 88.5)	0.50 (0.43, 0.56) 0.48 (0.41, 0.54)	0.96 (0.54, 1.70) 0.82 (0.48, 1.39)	0.36
12 months	Khorana score Threshold of 2 points Threshold of 3 points	21.1% vs. 15.1%, *p* = 0.06 18.8% vs. 18.1%, *p* = 0.87	60.6 (50.5, 70.0) 18.3 (11.4, 27.1)	49.6 (44.9, 54.2) 82.4 (78.6, 85.8)	21.1 (16.6, 26.2) 18.8 (11.7, 27.8)	84.9 (80.1, 89.0) 81.9 (78.1, 85.3)	0.55 (0.50, 0.60) 0.50 (0.46, 0.54)	1.51 (0.98, 2.32) 1.05 (0.61, 1.81)	0.10
	PROTECHET score Threshold of 2 points Threshold of 3 points	18.4% vs. 15.9%, *p* = 0.68 20.1% vs. 16.8%, *p* = 0.30	93.3 (86.6, 97.3) 48.1 (38.2, 58.1)	7.9 (5.7, 10.8) 57.5 (52.9, 62.0)	18.4 (15.2, 22.0) 20.2 (15.4, 25.7)	84.1 (69.9, 93.4) 83.2 (78.7, 87.1)	0.51 (0.48, 0.53) 0.53 (0.48, 0.58)	1.20 (0.53, 2.70) 1.25 (0.82, 1.92)	0.12
	CONKO score Threshold of 2 points Threshold of 3 points	20.8% vs. 11.0%, *p* = 0.01 20.0% vs. 15.1%, *p* = 0.28	68.3 (55.0, 79.7) 28.3 (17.5, 41.4)	49.7 (44.0, 55.4) 78.1 (73.0, 82.5)	20.8 (15.4, 27.2) 20.0 (12.1, 30.1)	89.0 (83.4, 93.3) 84.9 (80.2, 88.9)	0.59 (0.52, 0.66) 0.53 (0.47, 0.59)	2.13 (1.19, 3.81) 1.41 (0.76, 2.61)	0.17
	COMPASS-CAT score Threshold of 7 points Threshold of 11 points	19.1% vs. 19.1%, *p* = 0.99 18.3% vs. 20.0%, *p* = 0.65	69.5 (58.4, 79.2) 48.8 (37.6, 60.1)	30.5 (25.7, 35.7) 48.4 (43.0, 53.8)	19.1 (14.8, 24.1) 18.3 (13.4, 24.0)	80.9 (73.1, 87.3) 80.0 (73.9, 85.2)	0.50 (0.45, 0.56) 0.49 (0.43, 0.55)	1.00 (0.60, 1.68) 0.89 (0.55, 1.44)	0.25

## Data Availability

The data that support the findings of this study are available on request from the corresponding author. The data are not publicly available due to privacy or ethical restrictions.

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
