# Peer review of "External Validation of Risk Scores for Predicting Venous Thromboembolism in Ambulatory Patients with Lung Cancer"

_cancers, 2024, doi:10.3390/cancers16183165_

Round 1

Reviewer 1 Report

Comments and Suggestions for Authors

I found the article well written and interesting. The main question addressed by the research is the external validation of risk scores. Two of them were validated. I consider the results relevant for this field; however they aren't so original because the risk scores are well known. The paper addresses the performance in discriminatory capability of the Khorana score at a 2-point threshold  vs 3-point threshold and of  the CONKO score at 3-point threshold vs 2-point threshold. It could be important to use in clinical practice only few validated scores to decide about the start of prophylactic anticoagulation in patients with lung cancer. The paper presents data of external validation even if it is a retrospective study with a consiste number of missing data. The main problem was the retrospective design of the study. Apart from this, I'm not able to suggest any specific improvements. A sensitivity analysis could be considered by the authors. I think that the conclusions are consistent with the evidence presented and all the mai questions posed were addressed. In particular, Kaplan Meier Curves are effective to show that none of the analyzed scores was able to keep its discriminatory capability during the follow-up period. The references are appropriate. I suggest adding the articles that describe for the first time each score analyzed in the article. I think that Figure 1 is too long and I suggest to create a new one presenting the clinical outcomes.

I suggest a minor revision of the manuscript in order to publish it:

I suggest to change the phrase on lines 55-56 as following: “Having a VTE may require clinicians to temporarily stop the anticancer therapy and an increased risk of mortality.”

On line 64 I suggest to cute “(another DOAC)”.

On lines 61-64 I think that the authors would present the data not only as relative risk, but also as absolute risks.

On  line 153 I think that “Table 1. This is a table. Tables should be placed in the main text near to the first time they are cited.” is an error.

In my opinion Table 1 is too long; I suggest to divide it in two parts: “Patient-related risk factors”, “Medications during follow-up”, Cancer-related risk factors” and “Biomarkers” in a table and  “Clinical outcomes “ (not “other clinical outcomes) and “Risk scores” in another table.

One lines 238-239 I suggest to check this sentence, which would be more prudent.

On line 255 I think that it has to add “with prospective studies and larger samples”

Finally, I suggest to report, also as supplementary file, the scores cited in the test.

Author Response

Comments 1: I found the article well written and interesting. The main question addressed by the research is the external validation of risk scores. Two of them were validated. I consider the results relevant for this field; however they aren't so original because the risk scores are well known. The paper addresses the performance in discriminatory capability of the Khorana score at a 2-point threshold  vs 3-point threshold and of  the CONKO score at 3-point threshold vs 2-point threshold. It could be important to use in clinical practice only few validated scores to decide about the start of prophylactic anticoagulation in patients with lung cancer. The paper presents data of external validation even if it is a retrospective study with a consiste number of missing data. The main problem was the retrospective design of the study. Apart from this, I'm not able to suggest any specific improvements. A sensitivity analysis could be considered by the authors. I think that the conclusions are consistent with the evidence presented and all the mai questions posed were addressed. In particular, Kaplan Meier Curves are effective to show that none of the analyzed scores was able to keep its discriminatory capability during the follow-up period. The references are appropriate. I suggest adding the articles that describe for the first time each score analyzed in the article.?   I think that Figure 1 is too long and I suggest to create a new one presenting the clinical outcomes.

Response 1: Thanks for your invaluable comments and suggestions.

Firstly, we identified a coding error in analysis in that we had used continuous Eastern Cooperative Oncology Group Performance Scores (ECOG PS) (value ranges 0-4) rather than the expected binary indicators (i. e., whether or not the patients’ scores for ECOG PS were ≥2 points). This issue had resulted in an overestimation of CONKO scores. After correcting this issue, the CONKO score still showed a significant relationship with the VTE risk, but with the cut-off value of 2 points not 3 points.

Secondly, in the validation of the CONKO score, the number of missing data were considerable and the sample size for the analyses was smaller (n=370 with VTE events=60), compared with the sample size for validation of the Khorana score (n=570, VTE events=104) excluding those 90 patients who were lost to follow-up. The main cause of missing data was the unavailability of ECOG PS, while the CONKO score was modified from the Khorana score by replacing BMI≥35 kg/m2with ECOG PS.  If the individuals had a Khorana score excluding the predictor BMI≥35 kg/m2 was 2 or more points, they would have a CONKO score of 2 or more points regardless of the ECOG PS. Therefore, these patients could be included in the analysis of the CONKO score with the cut-off of 2 points to reach a bigger sample size (n=475, VTE events=83). This sensitivity analysis showed a similar result for CONKO score with the cut-off of 2 points had a statistically significant association with high VTE risk. This may have an important use in clinical practice. When the Khorana score cannot be calculated, e.g. blood tests are not done, the CONKO score could be an alternative assessment of high VTE risk if ECOG PS equals or higher than 2 points (see lines 185-191 and 204-207 in the clean version of major revision).

The articles which originally reported the development of each score were already cited (see References 10, 14-16 in the clean version of major revision). A supplementary file of the information of these scores has now been provided as per your suggestion.

Table 1 was split into two tables (Table 1 and Table 2) and the previous Table 2 is now Table 3. The manuscript was revised to reflect these changes (see pages 4-7 in the clean version of major revision).

Comments 2: I suggest a minor revision of the manuscript in order to publish it:

I suggest to change the phrase on lines 55-56 as following: “Having a VTE may require clinicians to temporarily stop the anticancer therapy and an increased risk of mortality.”

Response 2: Thanks for your edit. The manuscript has been revised with the addition of the suggested sentence at the end (see line 58 in the clean version of major revision).

Comments 3: On line 64 I suggest to cute “(another DOAC)”.

Thanks for your suggestion. We decided to remove “(another DOAC)” from the sentence to improve it (see line 66 in the clean version of major revision).

Comments 4: On lines 61-64 I think that the authors would present the data not only as relative risk, but also as absolute risks.

Response 4: The manuscript was revised accordingly to address this comment (see lines 62-66 in the clean version of major revision).

Comments 5: On  line 153 I think that “Table 1. This is a table. Tables should be placed in the main text near to the first time they are cited.” is an error.

Response 5: Thanks for picking up the error. It was corrected (see lines 159-160 in the clean version of major revision).

Comments 6: In my opinion Table 1 is too long; I suggest to divide it in two parts: “Patient-related risk factors”, “Medications during follow-up”, Cancer-related risk factors” and “Biomarkers” in a table and  “Clinical outcomes “ (not “other clinical outcomes) and “Risk scores” in another table.

Response 6: The tables were revised accordingly (see pages 4-6 in the clean version of major revision).

Comments 7: One lines 238-239 I suggest to check this sentence, which would be more prudent.

Response 7: This sentence was revised accordingly (see lines 258-259 in the clean version of major revision).

Comments 8: On line 255 I think that it has to add “with prospective studies and larger samples”

Response 8: It was added (see lines 273-274 in the clean version of major revision).

Comments 9: Finally, I suggest to report, also as supplementary file, the scores cited in the test.

Response 9: A supplementary file including the information about the four risk scores (the Khorana, PROTECHT, CONKO, and COMPASS-CAT scores) was added.

Reviewer 2 Report

Comments and Suggestions for Authors

THE LIMITATIONS OF THE STUDY ARE QUITE SERIOUS SO MY SUGGESTION IS OR TO CONSIDER YOUR WORK AS A PRELIMINARY RESEARCH,OR CONTINUE THE RESEARCH AND PRESENT A FUTURE "LESS- LIMITATION" WORK.THE STRUCTURE,THE IDEA,THE DATA ARE MORE THAN GOOD,BUT YOU CANNOT SUPPORT THE CONCLUSIONS WITH THE LIMITATIONS YOU REFEAR.KEEP IN MIND THAT THE REASON WHY WE HAVE ALL THESE SCORES IS THE DIFFICULTY TO FIND THE PROPER PROGNOSTIC FACTORS TO PUT IN THE EQUISION.

Author Response

Comments 1: The limitations of the study are quite serious so my suggestion is or to consider your work as a preliminary research,or continue the research and present a future "less- limitation" work.the structure,the idea,the data are more than good,but you cannot support the conclusions with the limitations you refear.keep in mind that the reason why we have all these scores is the difficulty to find the proper prognostic factors to put in the equision.

Response 1: Thanks for your comments and suggestions. We acknowledge the limitations of our research because of its retrospective nature However, according to the the Prediction model Risk Of Bias ASessment Tool (PROBAST), external validation of risk models needs at least 100 events. With the estimated VTE incidence of 12% (95%CI 9%-15%), more than one thousand participants with lung cancer are needed. Please notice that most published validation studies are commonly retrospective studies, like our study (e. g., Overvad, T.F., et al., Validation of the Khorana score for predicting venous thromboembolism in 40 218 patients with cancer initiating chemotherapy. Blood advances, 2022. 6(10): p. 2967-2976; and Van Es, N., et al., The Khorana score for prediction of venous thromboembolism in cancer patients: An individual patient data meta-analysis. Journal of thrombosis and haemostasis, 2020. 18(8): p. 1940-1951). Despite the limitations or our study (which have been explicitly mentioned in the manuscript), our original research highlighted the usefulness of two risk scores for patients with lung cancer. It also highlighted the over-time trend of decreased performance of the risk scores and the potential biomarkers, for example, the albumin levels, which could be a potentially included in the development of new risk scores or for updating current scores (i. e., recommendations for future research).

Reviewer 3 Report

Comments and Suggestions for Authors

Review of the paper: “External validation of risk scores for predicting venous throm-2 boembolism in ambulatory patients with lung cancer” by Ann-Rong Yan et al. 

Thank you for giving me the chance to read and review this interesting paper. 

The study focuses on a very selected topic like the specific VTE predicting scores in NSCLC patients is. 

The authors concentrated on the Korana and the CONKO scores, performing an external validation analysis using a series of 591 patients. Within this group of patients, they experienced VTE in 108 patients. 

Since this is a retrospective study, actually I don’t’ understand how the diagnosis was made; I fell this point should be clearer in the paper. 

English is good and the paper is well written. 

Author Response

Comments 1: Thank you for giving me the chance to read and review this interesting paper. 

The study focuses on a very selected topic like the specific VTE predicting scores in NSCLC patients is. 

The authors concentrated on the Korana and the CONKO scores, performing an external validation analysis using a series of 591 patients. Within this group of patients, they experienced VTE in 108 patients. 

Since this is a retrospective study, actually I don’t’ understand how the diagnosis was made; I fell this point should be clearer in the paper. 

Response 1: Thanks for your suggestion. VTE diagnoses were confirmed by medical imaging reports of Doppler ultrasound, CT pulmonary angiograms or on CT thorax done for staging or restaging purposes, which were documented in medical records. The manuscript was revised to address this matter (see lines 103-106 in the clean version of major revision).

Comments 2: English is good and the paper is well written. 

Response 2: Thanks for your comments.

Round 2

Reviewer 2 Report

Comments and Suggestions for Authors

ok.